

# Dry season diet composition of four-horned antelope *Tetracerus quadricornis* in tropical dry deciduous forests, Nepal

Chet Bahadur Oli[1], Saroj Panthi[2,3], Naresh Subedi[4], Gagan Ale[5], Ganesh Pant[1], Gopal Khanal[2,6,7] and Suman Bhattarai[8]

[1] Ministry of Forests and Environment, Department of National Parks and Wildlife Conservation, Babarmahal, Kathmandu, Nepal
[2] Ministry of Forests and Environment, Department of Forests, Babarmahal, Kathmandu, Nepal
[3] Faculty of Geo-Information Science and Earth Observation (ITC), University of Twente, Enschede, The Netherlands
[4] National Trust for Nature Conservation, Nepal
[5] Tribhuvan University, Central Department of Environmental Science, Nepal
[6] Post-Graduate Programme in Wildlife Biology & Conservation, Wildlife Conservation Society, India Program, National Centre for Biological Sciences, GKVK Campus, Bangalore, India
[7] Centre for Ecological Studies, Lalitpur, Nepal
[8] Institute of Forestry, Tribhuvan University, Pokhara, Nepal

Corresponding author
Saroj Panthi, mountsaroj@gmail.com

## ABSTRACT

It is essential to assess the feeding strategies of threatened species during resource-scarce seasons to understand their dietary niche breadth and inform appropriate habitat management measures. In this study, we examined the diet composition of four-horned antelope (FHA) *Tetracerus* and *quadricornis*, one of the least studied ungulate species, in Banke National Park, Nepal. A total of 53 fresh pellet groups were collected between December 2015 and January 2016 and analyzed using micro-histological fecal analysis technique. First, we prepared 133 micro-histological photographs of different parts of 64 reference plant species. Then we compared 1,590 fragments of 53 fecal samples with photographs of reference plants to assess the percentage of occurrence of different plant species in FHA diet. A total of 30 plant species belonging to 18 different families were identified in fecal samples. Chi-square goodness of fit tests showed that FHA appeared not to feed all plant uniformly. Out of 1,520 identified fragments in fecal samples, 1,300 were browse species and 220 were grass species. Browse represented 85.5% of the identified plant fragments, suggesting that FHA might be adopting a browser strategy at least during winter when grasses are low in abundance and their nutritive quality is poor. Tree species had the highest contribution in the diet (46.55%) followed by shrubs (24.52%). The family Gramineae was consumed in the highest proportion (27.68%) followed by Euphorbiaceae (11.95%). Overall, our results suggest that FHA has the feeding plasticity to adapt to resource fluctuation. Based on the findings of this study, we recommend that dicot plant species—particularly fruit trees and shrubs, which are the major source of nutrients for FHA during resource-lean, dry season—be conserved and natural regeneration of these taxa be promoted.

# INTRODUCTION

Knowledge of the diet composition of endangered wildlife species is very important to understand foraging ecology and to devise conservation management actions for their long-term persistence (*Belovsky, 1997*; *Ahrestani et al., 2016*). Such knowledge is particularly important for ungulates in seasonal environments (*Parker, Barboza & Gillingham, 2009*) where resource availability is pulsed in summer and scarcity is particularly acute during the arid winter season (*Styles & Skinner, 1997*; *Ahrestani, Heitkönig & Prins, 2012*). This seasonal flux in quality and quantity of resource availability (e.g., forage) often has nutritional costs for ungulates (*Parker, Barboza & Gillingham, 2009*). For example, reduced availability of preferred forage has been found to alter the composition of graminoid and browse in the diet, negatively influencing the maintenance of body mass of American elk *Cervus elaphus* during winter (*Christianson & Creel, 2009*). In the Mediterranean region, hares *Lepus europaeus* were found to eat herbs (preferred food) in the wet season but increase their diet breadth in the dry season by consuming herbs, fruits, and grains (*Sokos, Andreadis & Papageorgiou, 2015*). In the Indian trans-Himalaya, a medium-sized ungulate grazer, the blue sheep bharal, (*Pseudois nayaur)* was found to have a mixed diet (mainly browse) during resource-limited winter seasons due to reduced availability of graminoids, resulting from competition with domestic livestock (*Mishra et al., 2004*; *Suryawanshi, Bhatnagar & Mishra, 2010*). Change in diet balance affects reproduction, growth, and survival of animal influencing life history parameters such as body mass of adult females which correlates with vital rates like birth mass, growth rates and survival of young (*Pekins, Smith & Mautz, 1998*). Understanding the diet composition of a species during resource-lean season is therefore critical to understand diet plasticity and inform forage management measures.

The four-horned antelope (FHA) *Tetracerus quadricornis* is a medium-sized, solitary ungulate (adult shoulder height 55–65 cm, weight 18–21 kg) endemic to the Indian sub-continent (*Leslie & Sharma, 2009*).  It is widely but patchily distributed with fragmented populations in dry deciduous forests from the Himalayan foothills in Nepal to the Gangetic floodplains and the Peninsular mainland in India (*Rahmani, 2001*; *IUCN SSC Antelope Specialist Group, 2017*). Estimates suggest that fewer than 10,000 FHA remain in the wild (*IUCN SSC Antelope Specialist Group, 2017*). However, the population of FHA is suspected to have declined throughout its range, mainly due to habitat loss and fragmentation (*Sharma, Rahmani & Chundawat, 2009*). Although presently it is classified under the 'Vulnerable' category, the assessment of the IUCN Red List of threatened species states that "no subpopulation is estimated to contain more than 1000 mature individuals and it is possible that it is already close to reaching the Endangered category" (*IUCN SSC Antelope Specialist Group, 2017*). In Nepal, FHA is reported to occur in dry deciduous hill sal *Shorea robusta* and mixed Shorea-Terminalia forests in four protected areas of Nepal: Bardia National Park (*Pokharel, 2010*; *Kunwar et al., 2016*), Chitwan National Park (*Pokharel, Ludwig & Storch, 2015*), Parsa National Park and Banke National Park (*DNPWC, 2017b*). Its distribution is restricted to open canopy dry deciduous mixed forests, characterized by short grassland patches, sparse understory and undulating terrain (*Krishna et al., 2009*;

*Sharma, Rahmani & Chundawat, 2009*; *Baskaran et al., 2011*). It has been found to be sympatric with barking deer *Muntiacus muntjak* in the monsoon season in Nepal (*Pokharel et al., 2015*). Nepal's National Parks and Wildlife Conservation Act, 1973 has listed this species under the protected species list, prohibiting hunting (*GoN, 1973*).

To date, studies on wild populations of FHA have been focused on its distribution (*Krishna, Krishnaswamy & Kumar, 2008*; *Sharma et al., 2013*; *Pokharel, Ludwig & Storch, 2015*) and habitat ecology (*Sharma, Rahmani & Chundawat, 2009*; *Baskaran et al., 2011*) with few studies on its feeding ecology (*Sharma, Rahmani & Chundawat, 2009*; *Baskaran et al., 2011*; *Pokharel et al., 2015*; *Kunwar et al., 2016*). Although these previous studies have been useful in improving our understanding of the natural history, ecology and behavior of the species, we still know little about the responses of the species to changes in habitat components, interspecific interaction with other sympatric species, habitat requirements and population abundance. Since it continues to lose its habitat to agricultural development, livestock grazing, fire, and encroachment by invasive species like Banmara (*Lantana camara)* (*Krishna et al., 2009*), information on diet composition is particularly important for conservation management interventions. Previous studies showed that FHA predominantly consumes a browse-dominated diet, especially with highly nutritious plant parts such as fruits, flowers and fresh leaves (*Baskaran et al., 2011*; *Pokharel et al., 2015*; *Kunwar et al., 2016*). In summer, when the availability of grass is high, FHA has been found to increase its diet breadth and consume grass species as well as the forb species *Ageratum conyzoides* (*Kunwar et al., 2016*). *Cynodon dactylon* and *Acacia nilotica* were identified as the main winter dietary species of FHA in Madhya Pradesh, India (*Sharma, Rahmani & Chundawat, 2009*). The browse to grass ratio was high in the dry winter season and low in the wet monsoon season in the diet of FHA in Bardia, Nepal (*Kunwar et al., 2016*).

While previous studies on food habits of FHA have provided important insights into its seasonal pattern of feeding revealing its generalized feeding strategy, more in-depth and rigorous studies are needed to confirm if the findings of these species are applicable to all habitat conditions. Most of the previous studies had a small sample size (e.g., 20 pellet samples for dry winter season feeding analysis; (*Kunwar et al., 2016*)) making it difficult to draw any broad generalization of their diet patterns. Studies with sufficient sample size are needed not only to understand the variability present in the diet but also to ensure the validity of broader inferences. It has been documented that an ungulate species may be forced to consume different food species in different sites due to difference in food density and composition as well as the density of other co-occurring species, habitat, predation risk, monsoon seasonality and competition with sympatric species including livestock (*Fritz, Garine-Wichatitsky & Letessier, 1996*; *Wilsey, 1996*; *Valeix et al., 2009*). Site-specific studies on diet composition can thus be very useful not only in informing site-specific habitat management and species conservation measures but also in improving our understanding of the species feeding ecology in diverse habitat types and developing a general theory. Banke National Park, which lies in the foothills of the Siwalik mountain range, has diverse habitat types from pure *Shorea robusta* forests to mixed dry deciduous Shorea-Terminalia-Albizzia forests. Before it was established as a national park in 2010, it was managed as a production-forest to produce timber and fuel wood. Livestock grazing

and human use of the landscape for the collection of fodder and non-timber forests products was also common under previous management regime. The density of other sympatric ungulates (e.g., barking deer, spotted deer *Axis axis*) and the density of potential predators is less in comparison to other national parks where FHA occurs (e.g., Bardia National Park). These peculiarities offer a unique opportunity to assess if food habits of FHA in this national park are consistent with findings from other protected areas.

In this study, we examined the dietary composition of FHA in Banke National Park, Nepal, which is the first of its kind in this park. We specifically examined whether FHA consumes all potential forage plant species equally when the availability of such species is low. We hypothesized that if FHA is a selective browser, it would include a high proportion of browse in its diet. We also predicted that if this animal has a more flexible generalized grazer- browser mixed feeding strategy, it would continue to consume grasses despite their low quality in dry season while balancing the composition of dicots, which retain their nutritive quality during winter. The findings are useful for the government of Nepal and conservation stakeholders for planning forage and habitat management measures.

## MATERIALS AND METHODS

### Study area

This work was conducted with research permission (1082-2072-9-2) from Department of National Parks and Wildlife Conservation for research in Banke National Park (N27°58′13′ to N28°21′26″ latitude; and E81°39′29″ to E82°12′19″ longitude). This park extends along the Churia foothills of the western part of the Terai Arc Landscape of Nepal (Fig. 1). Established in 2010 as an effort to conserve the tropical deciduous ecosystem and to double the tiger *Panthera tigris* population in Nepal, it covers an area of 550 km$^2$ in its core zone and 343 km$^2$ in its buffer zone (*DNPWC, 2017a*). The park connects the Bardia National Park in the west and Suhelwa Wildlife Sanctuary of India through the forests in the southern part, with its buffer zone. Its elevation ranges between 153 to 1,247 m above the mean sea level. Mean maximum temperature is around 40 °C in summer but drops to very low during winter. Seasons are of four types, monsoon (Jun–Sep; the wet season with abundant rainfall), autumn (Oct–Nov), dry winter (Dec–Feb) and spring (Mar–May). The park contains eight ecosystem types: *Shorea robusta* forest, deciduous riverine forest, savannas and grasslands, mixed hardwood forest, floodplains, Bhabar and foothills of Chure range (*DNPWC, 2017a*).

### Data collection

Field surveys were conducted between December 2015 and January 2016 to collect the pellets of FHA and vegetation samples. Before going to the field for data collection, 22 key informant interviews were conducted with local people and park staff to identify the potential habitats of FHA. Based on information obtained from the key informant interview, we identified FHA hotspots and randomly laid transects of 500 m long and 20 m width on a map. Transect surveys are widely used method to collect fecal samples of ungulates (*Pokharel et al., 2015*; *Kunwar et al., 2016*). The survey team, which included the first author, three field assistants and an expert from National Trust for Nature

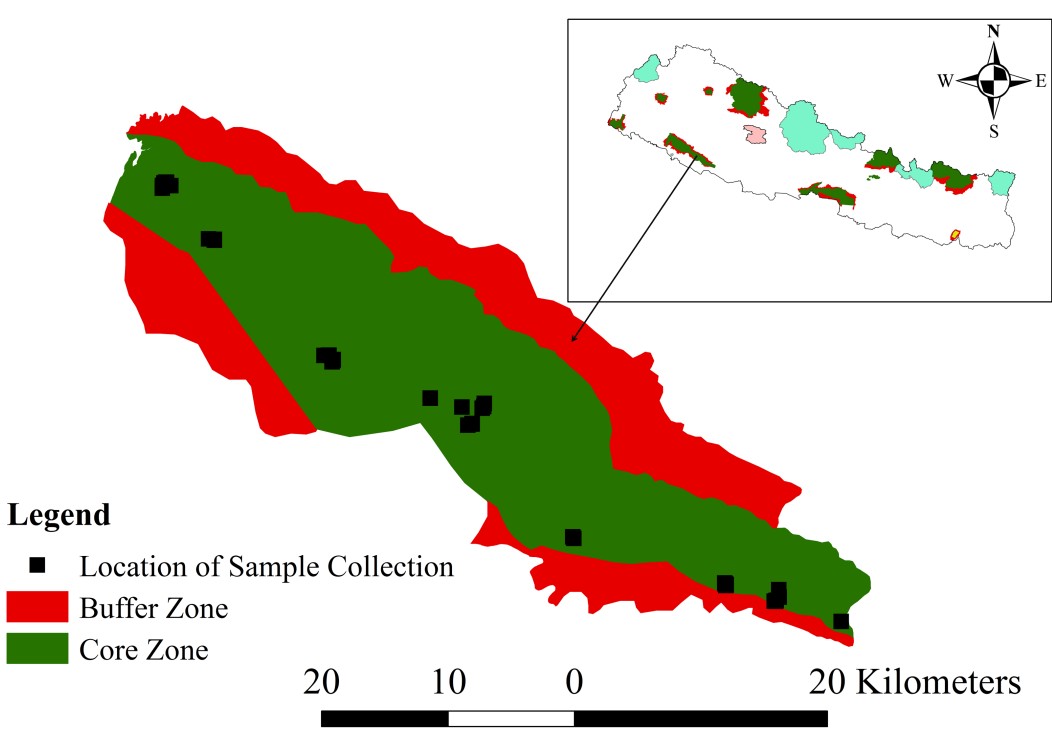

**Figure 1** **Map of the study area, Banke National Park, showing the core and buffer zones and the locations of sample collection.** The inset shows the location of Banke National Park within Nepal. Colored areas on the inset map indicate other protected areas (source of shape file: *UNEP-WCMC & IUCN, 2017*).

Conservation - Bardia Conservation Program walked along the 40 transects to collect the pellets samples. Wherever we recorded pellets, we established a plot of 10 m × 10 m around the pellet and collected the fecal samples and sample of all species of vegetation within these plots for lab analysis. This is a recommended and widely used plot size for the study of dietary patterns of wild animals (*Schemnitz, 1980*; *Panthi, 2011*; *Panthi et al., 2012*; *Aryal et al., 2015a*). Leaves, twigs, fruits, and barks of all plants were collected.

The pellets of FHA were identified checking the shape, size, and texture of pellets following *Pokharel (2010)* who has confirmed size and shape details of FHA pellets by installing camera traps in the suspected middens of FHA in Bardia National Park (see Fig. S1). These FHA pellets were available as a reference for the verification of the pellets at Bardia National Park. These reference pellets and the assistance of a trained wildlife technician (Mr. Binti Ram Tharu) from NTNC-BCP helped to minimize misidentification of pellets during the field survey. In drier habitat, the pellets can decay very rapidly, and further laboratory analysis can be difficult (*Jung & Kukka, 2016*) so fresh pellets, not more than seven days old, were identified based on texture and moisture content. We randomly sub-sampled 25 % each sample group for further analysis. These samples were air dried for five days in the field to remove moisture and prevent fungal growth. The collected plant samples were preserved in the herbarium and stored in the well ventilated dry room of

the Banke National Park Office, Overy Banke and sent to Central Department of Botany, Tribhuvan University, Kathmandu for further verification.

## Micro-histological analysis

Micro-histological fecal analysis technique was used to determine plant composition of FHA fecal matter (*Sparks & Malechek, 1968*; *Holechek & Gross, 1982*). This method is widely used as a diet analysis tool to investigate the dietary composition of ungulates (*Shrestha, Koirala & Wegge, 2005*; *Nagarkoti & Thapa, 2007*; *Aryal et al., 2015b*; *Jung, Stotyn & Czetwertynski, 2015*; *Wangchuk, Wegge & Sangay, 2016*). This method involves microscopic recognition of indigestible plant fragments of plant groups and preparation of reference and fecal slides and their interpretation. Samples of plant parts were dried in the oven at 60 °C in the laboratory and ground separately into powder using an electric blender. The powder of each sample was sieved using a 212 mesh.

The micro-histological slides of reference plants, as well as fecal sample slides, were prepared using the methods of *Norbury (1988)*. In this method, reference samples or fecal samples were placed in Petri dishes and bleached with 50 ml of 4% sodium hypochlorite for 6–24 h at room temperature to remove mesophyll tissue and to render the epidermis identifiable. The bleached contents were then rinsed well in a sieve, and then the rinsed fragments were stained with a few drops of a gentian violet solution (1 g/100 ml water) for 10 s and again well rinsed. The stained fragments were mounted on standard microscope slides in a DPX Mountant medium and covered with a cover slip (*Norbury, 1988*). Both reference slides and fecal pellet slides were observed immediately after preparation at magnification 400× with a digital microscope, and each fragment was auto-photographed using Bel Photonics (*Norbury, 1988*; *Panthi et al., 2015*). A diet analysis expert (Mr. Binod Shrestha) trained the first and fourth authors to identify the plant fragments. A total of 133 micro-histological photographs of different features of 64 plant species were prepared for the reference library. For each sample, 30 non-overlapping and distinguishable fragments were observed by moving the slides from left to right in the microscope. Specific histological features such as cell wall structure, shape and size of cells, trichomes; and shape and size of stomata were identified as key features to match the features of fecal plant fragments with reference plant (*Panthi, 2011*; *Aryal et al., 2012*).

## Data analysis

The plant fragments identified from the micro-histological analysis of the pellet samples were assigned into one of the following four levels of classification with different categories under each classification: (1) growth form: (i) grasses, (ii) forbs, (iii) shrubs, (iv) climbers (vine plants) and (v) trees; (2) class: (i) monocots and (ii) dicots; (3) family; and (4) species. The idea behind this classification was to assess the relative contribution of different categories of plant taxa under each classification to the diet of FHA. We added the total number of fragments of each species and rounded to the nearest 5 fragments.

Diet composition was expressed as the percentage occurrence of plant species (*Cavallini & Lovari, 1991*).

$$\text{Percentage Occurence} = \frac{\text{Number of fragments of a species or other category}}{\text{Total number of plant fragments identified}} \times 100$$

we performed the goodness of fit chi-square test to identify whether FHA ate all plants uniformly. Our research hypothesis was that FHA would not eat all plants species, family, growth form (grass, forb, climber, shrub, and tree) and class (monocot and dicot) uniformly. We also hypothesized that FHA would be a browser during winter. All tests were performed using Microsoft Excel and R software version 3.4.1 (*R Core Team, 2013*).

## RESULTS

A total of 1,590 plant fragments from 53 pellet samples were analyzed through micro-histological technique. Out of the total plant fragments, (4.4%) were unidentified, and these were excluded from statistical analysis. A total of 30 species belonging to 18 different families were identified in the pellets of FHA. Out of 30 species, the FHA diet included 14 tree species, eight shrubs, two forbs, five grasses, and one climber (Table 1). The dicot shrub species *Phyllanthus emblica* had the highest percentage occurrence in FHA diet (6.92%) whereas the dicot shrub *Clerodendrum viscosum* had the lowest percent occurrence (0.94%). FHA appeared not to feed all plant species uniformly ($\chi^2 = 312.56$, $df = 29$, $p < 0.001$) at the species level. Similarly, at the family level, FHA did not consume all plant families uniformly ($\chi^2 = 1982.41$, $df = 17$, $p < 0.001$). The family Gramineae which consists of 9 species contributed 27.68% of the diet whereas Verbenaceae contributed only 0.94% of the diet (Table 1). At the growth form level, FHA did not consume all growth forms (grass, forb, climber, shrub, and tree) uniformly ($\chi^2 = 1001.71$, $df = 4$, $p < 0.001$). In general, trees constituted a large proportion of diet contributing 46.55%, followed by shrubs (24.52%,), grasses (13.84%,), forbs (8.18%) and climber (2.52%) (Table 1).

Similarly, FHA did not use plants equally at the class (monocotyledonous and dicotyledonous) level ($\chi^2 = 229.01$, $df = 1$, $p < 0.001$). A total of 66.36% of FHA's diet was composed of dicotyledonous plants, and 29.25% of FHA's diet was monocotyledonous. The study identified 1,300 fragments of browse (forbs, climbers, shrubs, and trees) and 220 fragments of grass in FHA's diet. The ratio of browse to grass was found to be 85.53%: 14.47%, showing a strong affinity towards browse plant species in the dry season.

## DISCUSSION

Assessment of the dietary choices of a species during low resource availability period is critical to understand its foraging plasticity and inform subsequent habitat and forage management measures. In this study, we examined the winter season food habit of FHA, a sparsely distributed yet threatened species native to Nepal and India (*IUCN SSC Antelope Specialist Group, 2017*), based on micro-histological analysis of the collected fecal pellet samples. We hypothesized that if FHA is a selective browser during winter, it should show evidence of selectively foraging on browse in its diet.

Our result shows that dicots had a significantly higher percentage of occurrences in FHA pellets than monocots (suggesting that FHA might be adopting a browser strategy at least during winter when graminoids and grass species are low in abundance). Plant species differ in protein and fiber contents which influences animals' food choice (*Klaus-Hügi et al., 1999*). Smaller antelopes have smaller stomach compared to larger ruminants but

**Table 1** Percentage compositions of various plant categories identified in pellets of FHA.

| Family | Species | Class | Growth form | Percent occurrence |
|---|---|---|---|---|
| Gramineae | *Hemarthria compressa* | Monocot | Forb | 6.29 |
| | *Imperata cylindrica* | Monocot | Grass | 4.09 |
| | *Eulaliopsis binata* | Monocot | Grass | 3.14 |
| | *Bambusa vulgare* | Monocot | Tree | 2.83 |
| | *Thysanolaena maxima* | Monocot | Shrub | 2.83 |
| | *Themeda triandra* | Monocot | Grass | 2.52 |
| | *Heteropogon contortus* | Monocot | Grass | 2.2 |
| | *Cynodon dactylon* | Monocot | Forb | 1.89 |
| | *Digitaria* spp. | Monocot | Grass | 1.89 |
| Gramineae total | | | | 27.68 |
| Compositae | *Terminalia alata* | Dicot | Tree | 4.4 |
| | *Terminalia chebula* | Dicot | Tree | 2.52 |
| | *Terminalia belerica* | Dicot | Tree | 1.57 |
| Compositae total | | | | 8.49 |
| Euphorbiaceae | *Phyllanthus emblica* | Dicot | Shrub | 6.92 |
| | *Mallotus philippensis* | Dicot | Tree | 5.03 |
| Euphorbiaceae total | | | | 11.95 |
| Leguminoseae | *Acacia catechu* | Dicot | Tree | 4.72 |
| | *Bauhinia vahlii* | Dicot | Climber | 2.52 |
| Leguminoseae total | | | | 7.24 |
| Rubiceae | *Xeromphis spinosa* | Dicot | Tree | 5.97 |
| Rhamnaceae | *Zizyphus mauritiana* | Dicot | Tree | 4.4 |
| Oleaceae | *Nyctanthes arbortristis* | Dicot | Shrub | 3.77 |
| Apocynaceae | *Carissa spinarum* | Dicot | Shrub | 3.46 |
| Dipteriocarpaceae | *Shorea robusta* | Dicot | Tree | 3.46 |
| Lythraceae | *Woodfordia fruiticosa* | Dicot | Shrub | 2.83 |
| Anacardiaceae | *Buchanania lanzans* | Dicot | Tree | 2.52 |
| Myrtaceae | *Eugenia* spp. | Dicot | Tree | 2.52 |
| Sapindaceae | *Schleichera oleosa* | Dicot | Tree | 2.52 |
| Rutaceae | *Aegle marmelos* | Dicot | Tree | 2.2 |
| Tilaceae | *Grewia* spp. | Dicot | Shrub | 2.2 |
| Myrsinaceae | *Myrsine semiserrata* | Dicot | Tree | 1.89 |
| Liliaceae | *Asparagus phillipensis* | Monocot | Shrub | 1.57 |
| Verbenaceae | *Clerodendrum viscosum* | Dicot | Shrub | 0.94 |
| Unidentified | | | | 4.4 |
| Identified total | | | | 95.6 |
| Dicot total | | | | 66.36 |
| Monocot total | | | | 29.25 |
| Tree total | | | | 46.55 |
| Shrub total | | | | 24.52 |
| Grass total | | | | 13.84 |
| Forb total | | | | 8.18 |
| Total | | | | 100 |

have high metabolic requirements. This prohibits them from feeding large quantities of coarse grass species that are high in fiber and low in protein (*Owen-Smith, 1992*). In dry deciduous tropical forests, graminoids lose their palatability and nutritive quality during the dry season in comparison to wet season (*Sukumar, 1989*; *Baskaran, 1998*). This could probably explain why monocots were not eaten as much as dicots. *Berwick (1974)* and *Sharma, Rahmani & Chundawat (2009)* concluded that FHA is a selective feeder. The food selectivity by FHA may result from nutritional requirements; they need to decrease fiber intake, and maximize protein intake in order to increase digestibility.

Our results support the hypothesis that FHA adopts a browser strategy during winter, but we cannot rule out the possibility that FHA is a mixed feeder with substantial feeding plasticity to balance nutritional requirements. The presence of grasses in 14.3% of plant fragments suggests that grasses also have a substantial contribution to FHA diet. Our results of higher contribution of browse are consistent with the findings of *Kunwar et al. (2016)* who reported that browse constituted nearly two-thirds (66.95%) of the overall diet while grass species occurred only 13.68% (the rest, 19.77% remained unidentified). A study from India has, however, shown that FHA had more or less equal proportion of grass and browse in FHA diet in the winter season (14 grass, five herbs, four trees and one shrub) (*Baskaran et al., 2011*). This discrepancy in findings could be due to differences in study location, sample size and the high proportion of unidentified plants in their analysis. *Baskaran et al. (2011)* had 48% of the plant remains in their FHA fecal samples which could not be identified whereas in our study we have only 4.40% of the plant fragments that remained unidentified.

Our results showed plant species differ significantly in their contribution to FHA diet (Table 1). The shrub *Phyllanthus emblica* of the family Euphorbiaceae occurred most frequently (6.92%) in FHA diet. In their study in Bardia National Park, *Kunwar et al. (2016)* identified *Berlaria cristata* as the shrub species with the highest frequency of occurrence (5.33% of total fragments identified) in FHA diet in the winter season. The cafeteria experiments of *Berwick (1974)* in Gir forest ecosystem, India, and *Sharma, Rahmani & Chundawat (2009)* in Van Vihar National Park cum Zoo in Bhopal, India, showed that *Zizyphus mauritiana* contributed most to the diet of captive FHAs in winter. Our study also revealed a moderate contribution (4.40%) of *Zizyphus mauritiana*. Although *Zizyphus mauritiana* is highly palatable, its thorns inhibit its consumption in the natural habitats (*Berwick, 1974*). The FHAs in the Banke National Park do not appear to use many plants of the climber growth form as indicted relatively low percentage of occurrence in fecal samples.

FHA distribution is determined by the tree species richness in India (*Sharma, Rahmani & Chundawat, 2009*). In our study, tree species constituted a substantial proportion of FHA diet. On the whole, trees contributed the highest proportion (46.54%) of diets of FHA followed by shrubs (24.53%), grasses (13.84%), forbs (8.18%) and climbers (2.52%). But *Baskaran et al. (2011)* showed in tropical forests of southern India during the dry season that grasses were the major constituent of FHA diet (28.6%) followed by trees (8.0%), shrubs (5.6%) and herbs (6.7%). Our findings of the higher proportion of browse in FHA's diet supports the results of the feeding observations made on this species in Bardia National

Park, Nepal (*Kunwar et al., 2016*) and captive antelopes in India (*Solanki & Naik, 1998*). Our results also show the high proportion of the Gramineae family in the diet of this species similar to the findings of *Kunwar et al. (2016)*. Although *Baskaran et al. (2011)* assert that FHA is the generalist in feeding strategy, our study showed that it consumes more browse plant species than grasses in the winter season. According to *Hofmann (1989)*, concentrate feeders choose a high quality diet and show a remarkable degree of forage selectivity. Some herbivores such as elephants graze in the monsoon season and browse in the winter season (*Pradhan et al., 2008*). Our results show that FHAs in Banke National Park may have the plasticity to behave as concentrate feeders, consuming different proportions of various plant species and growth form.

During the monsoon season grass availability is high so the ungulates behave more like pure grazers because they can find palatable grasses everywhere, but they behave more like browsers in winter, a season of resource scarcity (*Pradhan et al., 2008*). Consistent with that finding, we found the FHA to act as a browser in resource scarce seasons. Browse was the major contributor to FHA's diet in all seasons, but the proportion of trees in the diet was high in the winter season and low in summer and monsoon season (*Kunwar et al., 2016*). Similarly, we found a high browse to grass ratio in winter season.

The micro-histological analysis method which we used for our study, includes multiple successive sampling from the individuals, pellets and epidermis fragments. Sample size, therefore, could affect the estimates of species diversity in the diet (*Katona & Altbäcker, 2002*). In our study, we randomly read 30 plant fragments per slide per pellet from 53 independent pellet groups for determining FHA diet which we hope provides a reasonable sample size. Of the total plant fragments, only 4.40% diet remained unidentified in this study. This percentage was 48% in *Baskaran et al. (2011)*. *In-vitro* digestibility also greatly influenced the results of micro-histological analysis particularly in the estimation of grass and forb content (*Vavra & Holechek, 1980*). FHA eats fruits, flowers and fresh leaves (*Berwick, 1974*; *Baskaran et al., 2011*) which are highly digestible. Thus, this percentage of unidentified plants in the diet could be due to high mastication and efficient digestion by the animal. We collected pellets and plants samples from only one protected area during a single season. More rigorous and detailed information could be obtained from multi-season and multi-site study.

Overall, our results suggest that FHA has the feeding plasticity to adapt to resource fluctuations. Future studies on nutrient content analysis of different diet plant species and causes of changes in diet composition across seasons would be particularly useful for habitat conservation and management. Based on the findings of this study, we recommend that dicots, particularly fruit trees and shrubs, which are the major source of nutrients for FHA especially during winter, be conserved and natural regeneration be promoted.

## ACKNOWLEDGEMENTS

We thank the Department of National Parks and Wildlife Conservation of Nepal and Banke National Park for providing research permission. We acknowledge Prof. Dr. Santosh Rayamajhi, (Institute of Forestry, Tribhuvan University, Nepal) and Prof. Dr. Tej Bahadur

Thapa (Central Department of Zoology, Tribhuvan University, Nepal) and Mr. Binod Shrestha for their guidance during the study and Central Department of Environmental Science, Tribhuvan University for providing the laboratory facility. We thank Dr. Hillary Young (Department of Ecology, Evolution, and Marine Biology, University of California, Santa Barbara, USA) for her contribution to improve the English language and other technical issues during the manuscript revision.

### Funding

This work was supported by the National Trust for Nature Conservation-Bardia Conservation Programme. The funders had no role in study design, data collection and analysis, decision to publish, or preparation of the manuscript.

### Grant Disclosures

The following grant information was disclosed by the authors:
National Trust for Nature Conservation-Bardia Conservation Programme.

### Competing Interests

The authors declare that there are no competing interests.

### Author Contributions

- Chet Bahadur Oli conceived and designed the experiments, performed the experiments, analyzed the data, contributed reagents/materials/analysis tools, prepared figures and/or tables, authored or reviewed drafts of the paper, approved the final draft.
- Saroj Panthi analyzed the data, contributed reagents/materials/analysis tools, prepared figures and/or tables, authored or reviewed drafts of the paper, approved the final draft.
- Naresh Subedi and Ganesh Pant authored or reviewed drafts of the paper, approved the final draft.
- Gagan Ale performed the experiments, authored or reviewed drafts of the paper, approved the final draft.
- Gopal Khanal contributed reagents/materials/analysis tools, authored or reviewed drafts of the paper, approved the final draft.
- Suman Bhattarai conceived and designed the experiments, performed the experiments, authored or reviewed drafts of the paper, approved the final draft.

### Field Study Permissions

The following information was supplied relating to field study approvals (i.e., approving body and any reference numbers):

Research permission (1082-2072-9-2) was granted from the Department of National Parks and Wildlife Conservation to conduct the study in Banke National Park.

### Data Availability

The raw data are provided in the Supplemental Files.

## Supplemental Information

Supplemental information for this article can be found online at http://dx.doi.org/10.7717/peerj.5102#supplemental-information.

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
