# Peer review of "Dry season diet composition of four-horned antelope Tetracerus quadricornis in tropical dry deciduous forests, Nepal"

_PeerJ, doi:10.7717/peerj.5102_

## Round 0.1 · original submission · Major Revisions

Although this manuscript is a fairly straightforward study of ungulate diet, making an editorial decision proved challenging. Reviewer 1 provided brief positive comments and indicated that the article was ready for publication. The PeerJ staff and I felt that it was necessary to seek an additional review because there was not enough detail in this reviewer's comments to be confident that he had given sufficient attention to all potential issues. Reviewer 2 raised several significant concerns and recommended major revision. Reviewer 3 also had important concerns and was more positive about the quality of the research in his comments but nevertheless recommended rejection. I believe that Reviewer 3's rejection recommendation derived from his view that the manuscript would be more appropriate as a short communication in a mammalogy journal. PeerJ's publication policy is that any contribution will be published, as long as the science is solid, without regard to its larger impact. My own reading of the manuscript indicated issues about the statistics and language. Based on the reviews and my evaluation, I believe that the manuscript requires major revisions and possibly a second round of reviews. In my comments below, I summarize my own concerns and emphasize some of the issues raised by the reviewers. It is important, however, that your revision considers all the reviewers' comments and provides a detailed response to each point.

Abstract
• I agree with the reviewer that the statistical values are not needed in the Abstract. In fact, the percentages of different categories in the diet and how they differ from summer values seem the most important result to emphasize. If the browse to grass ratio is based on number of species, you must make that clear. However, it is surprising that number of species rather than fragments in each category is the relevant issue.

Introduction
• Reviewers 2 and 3 both commented on needed changes in the Introduction.
• Following Reviewer 3, I suggest expanding the first paragraph to summarize well designed studies on other ungulates showing how diet changes in seasons of lower resource availability and how that information has been used to understand the ecology and conservation of these species.
• The second and third paragraph are not well organized, as the reviewers recognized. There is a mixture of information on distribution and habitat along with diet in both paragraphs.
• The second paragraph is an appropriate introduction to the species, but you should be sure that you are summarizing what is known over its range, not just Nepal or the park in which you worked as indicated by Reviewer 3. All diet information should be put into the next paragraph.
• The third paragraph should summarize what is already known about the diet. Both reviewers noted that your statements about diet are incomplete because additional information was introduced in the Discussion that contradicted your statement about the lack of previous study. Since your study is about winter, make it clear what information comes from summer and from winter, and lead clearly to the precise gap in knowledge that your study seeks to address. Any hypotheses you have about the diet and preference can be developed here. There seems to be a discrepancy between Introduction (L89) and Methods (L176) regarding your hypotheses.
• You need a new paragraph to address the ways in which diet can be addressed and their strengths and weaknesses. Because your study is also about preference, you need to address preference issues, why they are important and how they are studied in the same or an additional paragraph.
• Your final paragraph should develop your objectives in this study much more precisely, based on the research gaps and methods identified earlier in the Introduction. The order of objectives should match that in the Methods, Results and Discussion.

Methods
• The number of species and endangered species of vertebrates does not seem relevant to this study.
• Fig. 1 needs a complete caption.
• The purpose and procedure for the transect (L118) is unclear.
• The plant collection procedure is unclear. I can understand collecting samples for identification of plant species, but the protocol needs to be described, as noted by Reviewer 3. I do not see any information about quantitative sampling that would be required for assessing preference, so I assume that your use of chi square is to compare diet to a null hypothesis of equal contribution of all taxa. This needs to be clearer and you need to remove any indication that you are looking at preference.
• Reviewers 2 and 3 raised questions about how FHA pellets were identified. I see that you have provided some information, but it might be useful to list the other species that could be confused with them and how they were distinguished.
• I do not agree with Reviewer 2 that you need to provide the names of your assistants, but I think it would be useful to identify the expert who provided the pellet identifications.
• You do not mention any difficulties with the identification of the plant materials. Were there some related species that might have been confused? If so, how did you address this?
• Several of the articles you referred to indicated that considerable training was required before reliable identifications were possible. How did you address this issue?
• Reviewer 2 suggested providing the microhistological reference set as supplementary material. Looking at other recent research using this technique, I have not seen this done, and Reviewer 2 did not suggest it. I can see that it could be used to provide reference material for future studies. However, I leave the decision to you.
• The supplementary table on sample location is incomplete. It needs proper explanation with a complete caption and column headings that include the units. How should the reader interpret these numbers? Are the sites numbered in your set of samples?
• Your statistical tests need to be clarified and possibly revised.
o Did you use the raw numbers in your chi square? (Using proportions in a chi square test is not appropriate.)
o Check your calculations. I think that your df values are not correct.
o Did you check that your data meet the assumptions of chi square?
o How were the data for occurrence in feces from different samples and sites combined?
o Were there substantial differences in habitat between regions in the park? If so, would it be more appropriate to do separate analyses for regions with similar habitat?
o How were the data actually used to test the hypotheses you presented? In other words, what relationships in the data were expected if your hypotheses were correct and if they were not?

Results
o Because your study cannot assess preference since you did not measure availablity, you should be careful not to imply preference by significant chi squares in your results.
o You should explicitly relate your results back to any hypotheses that you presented.
o Table 1 is hard to understand because you centered the first three columns rather than putting them at the top of each section. It then is difficult to see which species fit in which category. I suggest putting the functional category, broad category, and family on the first line of each section.
o Your supplemental data set identical to Table 1. You should provide your detailed data as supplemental material. I think that you need to provide the number of fragments of each species found in each sample. Each sample needs to be identified by location, perhaps as an extension of the supplementary table that provides the location.

Discussion
o I found your Discussion not very clear and well organized. Be sure that each paragraph represents a distinct and coherent topic. Go back to the objectives and discuss your results in relation to those objectives. For example, when you compare your results to previous research, you do not make it clear which previous research is summer and which is winter.
o How do your findings relate to diet changes in relation to the active growing season in other ungulates?
o You should develop a more explicit discussion of the validity of your findings. Your mention of possible limitations such as digestibility is very brief. Yet, it seems that the literature on possible limitations of this method is quite extensive.
o Is it possible that the non-significant chi square for species but significant chi square at higher taxonomic levels could be a statistical artifact relating to a diverse diet?
o If an implication is that certain plant species should be protected, which ones should these be? What is the justification?

Language and style
o Overall, the manuscript is quite well written. However, the articles (a and the) are missing from many nouns, and there are other mistakes in word use and grammar. After completing the revisions, please have one or more readers fluent in English carefully check the text for such errors.
o Check the references carefully. There are spelling mistakes and inconsistencies such as capitalization of some journal article titles and lack of italics for scientific names.

·

Basic reporting

No comment. Nicely written article on a lesser known species

Experimental design

Original research work with well defined research question. Methods described with sufficient details.

Validity of the findings

Fairly good results. Conclusion are well stated, linked to original research questions.

Additional comments

Good work

Reviewer 2 ·

Basic reporting

no comments

Experimental design

no comments

Validity of the findings

na

Additional comments

1. Removed chi/statistical value from abstract make it, simple and understandable.
2. Introduction section is not well-organized need to with general background, species status, hypothesis, what are you testing for and why? And innovation in your research
3. Line 49: not clear about Species: animal or what ?? this paragraph not appropriate with starting background, add more about dietary stuff, line 55 is odd sentence does link above or later.. revised it.
4. It is not the first study as author claim more advanced study has been already done
Kunwar, A., R. Gaire, K.P. Pokharel, S. Baral & T.B. Thapa (2016). Diet of the Four-horned Antelope Tetracerus quadricornis (De Blainville, 1816) in the Churia Hills of Nepal. Journal of Threatened Taxa 8(5): 8745–8755; http://dx.doi.org/10.11609/jott.1818.8.5.8745-8755
Pokharel, K. P., Yohannes, E., Salvarina, I., & Storch, I. (2015). Isotopic evidence for dietary niche overlap between barking deer and four-horned antelope in Nepal. Journal of Biological Research-Thessaloniki, 22(1), 6.
4. Line 58 jumping: write about species background then distribution behaviours, diet and what has been done, problems, research gap etc..
5. Line 115 Key informant interviews what purpose what authors asked, whey, how many.
6. Line 117; how transect line was developed, how many, why they did survey in transect line,
7. Line 119 FHA pellets: how experts distinguish it??
8. Line 116 to 118: who were assistants and who were experts what the role was from authors on field survey???
9. Line:134 need to put the supplementary file with GPS points pellet presence sites.
10. Line 142: use the latest reference for this methodology:
Aryal A, SCP Coogan, W Ji, JM Rothman, D Raubenheimer. 2015. Foods, macronutrients and fibre in the diet of blue sheep (Psuedois nayaur) in the Annapurna Conservation Area of Nepal. Ecology and Evolution 5(18): 4006-4017.
Aryal A., D Brunton, W Ji, J Rothman, SCP Coogan, B Adhikari, S Juhnu, D Raubenheimer. 2015. Habitat, diet, macronutrient, and fiber balance of Himalayan marmot (Marmota himalayana) in the Central Himalaya, Nepal. Journal of Mammalogy 96(2): 308-316.
11. Line 149 to 154 add above reference
12. Line 160 A total of 133 micro-histological photographs Put this file and photo for as supplementary file for publication. These could be future reference for follow researcher..
13. In discussion need to discuss micro histological methods, its drawn back and how you address method bias and also discuss with this methods how it can use for herbivore and carnivores discuss it such as
Aryal A, J Hopkins, W Ji, D Raubenheimer, D Brunton. 2012. Distribution and diet of brown bear in the upper Mustang region, Nepal. Ursus 23 (2): 231–236.

Discussion section: discuss how non-invasive genetic analysis can be helpful for future research, conservation measure/effectiveness and impact of climate change etc… on this species for example:
Aryal A, D Brunton, T McCarthy, D Karmachharya, W Ji, R Bencini, D Raubenheimer. 2014. Multipronged strategy including genetic analysis for assessing conservation options for the snow leopard in the central Himalaya. Journal of Mammalogy 95(4):871-881.
Aryal A, UB Shrestha, W Ji, S Ale, T Ingty, T Maraseni, G Cockfield, D Raubenheimer. 2016. Predicting the distributions of predator (snow leopard) and prey (blue sheep) with climate change in the Himalaya. Ecology and Evolution 6(12):4065–4075.
discuss conservation issue and its preys species interaction and conservation measurement such as:
Aryal A, KP Acharya, UB Shrestha, M Dhakal, D Raubenhiemer, W Wright. 2017. Global lessons from successful rhinoceros conservation in Nepal. Conservation Biology. http://onlinelibrary.wiley.com/doi/10.1111/cobi.12894/full
Aryal A, RP Lamsal, W Ji, D Raubenheimer. 2016. Are there sufficient prey and protected in Nepal to sustain an increasing tiger population? Ethology Ecology and Evolution 28(1): 117-120.
Aryal A, W Ji, UB Shreshta, R Bencini, D Raubenheimer. 2015. Conservation conflict: Factor people into tiger conservation. Nature 522: 287-287.

·

Basic reporting

This manuscript is well organized and generally follows the scientific format; however, to be acceptable for publication it requires substantial revision in regards to the following:

1) The writing itself, while good, does need substantial work to improve the clarity and flow of the paper. Significant polishing of the text is necessary before being acceptable for publication. I provide some suggestions in the General Comments to the Author, but further polishing of the text will be required.

2) The Introduction wanders and does not really get at the purpose of the study and hypotheses/predictions tested by the authors. Much of the Introduction contains “filler” text about the species that is not particularly germane to the current study. In my view, the authors really need to focus on why understanding the winter diet of four-horned antelope is of interest. To do so, they need to really draw on the ungulate diet literature in general, and what is already known about FHA diets specifically. I was surprised that much of the reference on FHA diets was not mentioned earlier in the Introduction (but it was in the Discussion). This literature needs to be the backbone of the a revised Introduction, accompanied by a solid rationale and hypotheses (if applicable, see my comments below). Most importantly, the purpose of the study needs to be made crystal clear.

3) Too much of the text focuses on Banke National Park itself, FHA in Banke National Park, or studies on ungulates in Nepal. In my view, the authors need to focus the paper on diets of FHA while at the same time broaden the scope of the paper to ungulate diets in a global context. That is, I recommend that the authors a) delete much of the text about Banke National Park that has little to do with the current study; b) consider what is known about FHA diets and ecology across their range, not just in the park or Nepal; and c) draw much more heavily on the ungulate diet literature globally, than specific to Nepal.

4) There are many instances where I thought unnecessary information was provided by the authors that is not of the type that is normally provided in scientific papers (e.g., Lines 117, 123, 134, and others), and this information should be deleted.

5) The captions for Figure 1 and Table 1 need to provide much more detail about what information the figure and table provide. They should be able to stand alone.

6) Please iInclude both the common name and scientific name at first use for all species (e.g., barking deer at Line 70; tiger at Line 99, Hyena hyena at Line 107, and others).

7) Please avoid starting sentences with numbers.

Experimental design

I have difficulty evaluating the experimental design of this study because there really isn’t one there. In essence, this is a descriptive natural history study that doesn’t really aim to test any meaningful hypotheses. That is not a judgement on the value of the work, just to state that it is quite descriptive and not experimental.

As noted above in the Basic Reporting section. The research question is not well developed, and the context to assess whether it is relevant or meaningful, or fulfills an important information gap, is lacking. I want to believe that it is relevant and meaningful, but I need the authors to explicitly develop that argument for the reader; rather than having the reader trying to figure that out for themselves.

In terms of the study methods/techniques used to determine FHA diets, the authors do a good job of following standard protocols for microhistological analyses of ungulate diets from fecal samples. I have few substantive issues with how the work was done and, generally, it is reported to sufficient rigor to permit the study to be replicated. Good work.

Some methodological questions that should be addressed in a revised manuscript include:

1) How easily was it to determine FHA pellets from those of barking deer, or any other sympatric ungulate that also defecates pellets? It is quite important that the paper address the reliability in distinguishing between the pellets between FHA and sympatric species with fecal pellets that are similar.

2) If fresh pellets (within 7 days – Line 133) are important, then the authors need to address how they could ascertain this in the field, taking note that in arid environments pellets may dessicate at different rates in different habitats (see: Jung and Kukka. 2016. Wildlife Biology 22: 160-166; Hibert et al. 2010. European Journal of Wildlife Research 57: 495-503).

3) The authors need to describe in detail the sampling protocol for obtaining reference collections of available plants that FHA may use as forage.

Validity of the findings

The data from this study are robust and statistically sound – I have few substantive issues with the methodology employed.

A major issue that is only partially addressed in the manuscript is that the data come from a single year, at a single study site, during a single season. This imposes some major limitations on what can be inferred about the diet of FHA, and the authors need to tackle this limitation in the Discussion. For instance, how representative is this study area of the vegetation communities found across the range of FHA? Was the climate and weather typical the winter of the study, or might the availability of some of the forage resources vary from year to year, resulting in differences in use by FHA?

As noted above in the Experimental Design section, this is a descriptive study so the ability of the authors to link conclusions to supporting a priori hypotheses are limited.

Finally, a minor point, but the authors should discuss the limitations of microhistological anaylses. For example, some forage classes, such as forbs, are chronically underrepresented in microhistolocical analyses, which will bias the results.

Additional comments

Overall, I’d like to state that I enjoyed reading this manuscript and believe that it strives to fill an important gap in our knowledge of the ecological requirements of FHA. Thank you for doing this study.

I believe that through a critical and substantive edit to focus the manuscript on the question at hand – while more broadly drawing from the global literature – this study would be most appropriate as a Note or Short Communication in a mammalogy or zoology journal, such as Mammal Research, Mammal Study, Mammalia, or Mammalian Biology. The data is certainly valuable, but is limited in it’s spatial and temporal scope, which restricts making broader inferences about FHA diet.

I really hope the authors expend the effort to substantially rework this manuscript and publish it in an appropriate venue. The data are of interest and value, and the authors have done a lot of work in the field and lab to collect these data, which can benefit our understanding of FHA ecology.

Below are suggestions aimed at helping the authors improve their manuscript.

Introduction

Line 51: Replace “large ungulate herbivores” with “ungulates”

Line 52: Delete “such as alpine meadows, savanna grasslands, and dry forests of tropical regions”

Line 54: Replace “dry” with “the arid”

Line 55: Replace “have” with “has” and “on” with “for”

Line 56: Delete: the last part of this sentence, beginning at “, and thus studies on their…”

Line 58: Replace “one such solitary medium sized ungulate species, which patchily occurs” with “a medium-sized, solitary ungulate that is patichly distributed”

Line 60: Replace “animals of this species” with “FHA”

Line 62: Replace “decline” with “declined”

Lines 63-67: I think this long sentence about where they occur in Nepal is not relevant and should be deleted.

Line 68: So, if it is already known to be a browser based on other studies, I am wondering what, specifically, new information this study adds. The Introduction needs to explicitly layout the rationale and value of the present study, within the context of what is already known about the diet of FHA.

Methods

Line 96: This sentence should be in the Acknowledgements, not the Methods.

Lines 104-110: These lines aren’t needed and can be deleted. The paper is about FHA, not the park.

Line 115: I like that local knowledge was used to guide the field work. Well done.

Line 117: What defined the “expert”? Please explain.

Line 117: the number of field assistants is not important in the reporting of the work and should be deleted.

Line 119: Please describe or define, or provide a reference for “purposive sampling”, or delete this sentence.

Line 123: Delete the last part of this sentence, beginning with “and labeled in …”

Line 125: Rework this sentence using wording such as: “FHA pellets are usually elongated; however, in some cases, they may be cylindrical with a point on one end (Pokharel 2000).”

Lines 129-131: Delete the text here, beginning at “at the office of…” and ending at “pellets during field survey.” Unnecesary detail.

Line 131: This sentence, beginning with “ Total 53 pellets group…” belongs in the Results, not the Methods.

Line 134: Delete the last part of this sentence, beginning at “were collected in zip lock bags…” Unnecessary detail.

Line 135: Reword to “We randomly subsampled 25% of each pellet group for further analyses.”

Line 136: For how long were the samples air-dried?

Line 137: More importantly than where the collected plant materials stored, the authors need to describe in detail the sampling protocol for obtaining reference collections of available plants that FHA may use as forage.

Lines 143-145: Replace “of plant eating species in Nepal” with “ungulates”, and reduce the number of cited papers. I suggest that the focus be on cited works not only from Nepal but include the modern use of microhistological studies of ungulates from other parts of the world, in general (e.g., Jung et al. 2015. Journal of Wildlife Management 79:1277-1285; Jung. 2015. Mammal Research 60:385-391; Wangchuk et al. 2016. Journal of Natural History 50: 759-770; Cain et al. 2017. Wildlife Biology; ., or many, many others).

Line 149: Replace “through” with “using an”

Line 151: Replace “given by” with “of”

Line 170: What are “climbers”? Are these vines? Please define.

Line 175: Replace “eats” with “ate”

Line 178: Why was a P value of 0.01 used instead of the more widely used P > 0.05? I would think P > 0.05 would be sufficient.

Line 184: Replace “thirty” with “30”. Use numbers of it is ≥10.

Line 176: This is a fairly weak hypothesis. I suggest rephrasing to something along the lines of you tested for selection.

Results

Line 189: Here and elsewhere, please replace “<<” with “<”

Line 198: Here and elsewhere, please avoid beginning a sentence with a number.

Line 203: Add “the” after “species in”

Discussion

Line 220: These studies should be highligted in the Introduction as well, to provide context for the current study.

Lines 223-238: Same comment as for Line 220. These studies should be also discussed in the Introduction, to provide context for what is already known about the diet of this species.

Line 247: Replace “the tamed” with “captive”

Line 256: The statement that the study obtained a “reasonable optimum sample size” requires justification, or should be deleted. I would argue that 53 pellet groups is not a large sample from a statistical perspective; however, I appreciate the significant field effort that went into obtaining these samples, not to mention the amount of lab work to process this many samples. Good work.

---

## Round 0.2 · Major Revisions

I am returning your manuscript with a request for major revisions. I have not carried out a complete review because several issues that I raised in the previous review are still outstanding. It would be inefficient to review all your changes before those issues are addressed.

1) Validity of the chi square test
In the previous decision, I asked: Did you use the raw numbers in your chi square? (Using proportions in a chi square test is not appropriate.)
Your response: Among the three types of Chi-square test, we used goodness of fit. We have used proportion for Chi square test (data of table 1). We want to know whether FHA eats all dietary plants uniformly or not. So we are confident, using proportion in this case is correct. If you have any information to follow and to understand not to use the proportion, please recommend us that document we will follow your suggestion.

I spent some time online checking to see whether chi square could be done on proportions. All the formulas use (O - E) where O is observed frequency and E is expected frequency (not proportion). I can give an illustration of the problem using a hypothetical example. Suppose we have 4 categories (such as plant species) A,B,C and D and our hypothesis is that the number of samples (such as plant fragments) in each category will be the same. If we have 10 samples distributed such that there are 4 of type A, 3 of type B, 2 of type C and 1 of type D, the expected number in each category is 2.5. For this situation, the chi square value is 2, and with 3 df, p = 0.57 (nonsignificant). If we had 1,000 samples with the same proportional distribution (A 400, B 300, C 200, D 100), the chi square is 200 and the p < 0.0001 (highly significant). This makes sense because a small deviation from an expected distribution is more likely to be 'real' if the sample size is larger. If we had converted these values to percentages, both the would have come out to the same chi square of 20, p <0.0002. Therefore, the small sample size would have been interpreted as highly significant when it was not and the large sample size would have been interpreted as less significant than it really was. Therefore, it matters to use the actual numbers not the proportions. I did my calculations using an online source GraphPad/QuickCalcs https://www.graphpad.com/quickcalcs/chisquared2/. Note that I am not expert in statistics, so you can correct me if it is really clear than I am wrong. It might be a good idea to consult with a biostatistician to be sure that you are doing the test correctly.

In addition, you should indicate whether you included or excluded the category 'unknown' when you calculated the chi square (i.e., was 'unknown' considered as if it was a species or was it removed and you calculated chi square based on the real species? Note that if you remove the unknowns, sample size and therefore expected number in the chi square changes.) Also, if you excluded unknowns, then the total fragments from each sample would vary and the way you calculated the total would make a difference (see my next point).

2) Combining data
In the previous decision, I asked: How were the data for occurrence in feces from different samples and sites combined?
Your response: We can’t understand this comment. Please elaborate more. As far we know, you are asking about the fecal collection. We collected 53 fecal samples from 53 locations so they are different each other. Our aim was to know the dietary species. We also wanted to know whether all of these plant species were eaten uniformly. So don’t know about other things.

What I wanted you to specify was how the averages in Table 1 were calculated. For example, if the number of fragments differed between samples, it would make a difference to the final values if you calculated percent for each sample before you calculated the mean percent for each species or if you added the total number of fragments in each species and then calculated the total. Since your number of fragments is exactly 53 samples x 30 fragments per sample, the numbers should not be affected, but you should still state how the data were compiled.

3) Table 1
In looking at your data again, I notice that a reader is not able to find the percentages for any of your combined categories (families, broad taxonomic group, functional category). You provide some of this information in the text, but not all. I will provide you with an illustration of a way that you could revise Table 1 to make it more complete and informative as well as more compact. By ordering the species by families and then species, with the monocot families preceding the dicot families, the sums of each group are easily seen. Then, all you need is to list the growth form sums at the bottom of the list. You can use shading of cells that represent sums, so that they can be distinguished from individual species values. Also, please check your revised table carefully; there are spelling mistakes such as Gramineae, inconsistencies such as grass in the singular and the other growth forms in the plural, extra spaces between some lines and failure to align the number with the name for unidentified. Please align the percent column by the decimal. From my reading and consultation with a botanist, 'growth form' is a more appropriate term than 'functional category', and monocots and dicots seem to be called 'classes' rather than 'broad categories'. These changes would also apply to the Methods (L225-227). The final column was defined in your Methods as 'percentage of occurrence', which I think you could shorten to 'percent occurrence' in the table and methods. Diet composition is not appropriate as a column heading because the term refers to the combined result rather than the individual components. Please revise the heading to more accurately and completely describe the contents of the table, including the definitions of any abbreviations.

In proposing a revision to Table 1, I noticed that you state that there were 5 species of Gramineae whereas the table shows 9 species in this family. Please check all numerical data in your text to be sure that it agrees with the table.

4) Supplementary data
In the previous decision, I asked: Your supplemental data set identical to Table 1. You should provide your detailed data as supplemental material. I think that you need to provide the number of fragments of each species found in each sample. Each sample needs to be identified by location, perhaps as an extension of the supplementary table that provides the location.
Your response: We provided detail location of sample plots. This information is sufficient to know the location of sample collection.

You did not provide the supplementary information that I requested. The raw data needed for the supplementary table include the number of fragments of each plant species in each sample. As I implied in the previous request, this could be achieved by an extension of the sampling location table with a column for each plant species plus unknown and the number of fragments of each species in each sample. Please show the sums for each column and make sure that they agree with Table 1. This will help clarify the issues raised in point 2 above if a careful reader has questions. The supplementary data need to include in the heading any abbreviations used. The table name will also need to be changed to include the new information.

5) Training
In the previous decision, I asked: Several of the articles you referred to indicated that considerable training was required before reliable identifications were possible. How did you address this issue?
Your response: To identify the fecal sample, we had reliable sources (Pokhrel 2010) and expert trained by NTNC so we are confident that fecale sample we collected were pellets of FHA.

I was referring to the identification of plant fragments, not the origin of fecal pellets. It seems that identification of plant fragments is a skill that takes substantial practice to provide reliable results. I was asking you to indicate what training or other controls on accuracy you used in your study. This information is included in some of the articles you cited.

6) Browse to grass ratio
Are you sure that you have correctly calculated the browse to grass ratio? Some of your monocots are forbs, shrubs, and trees. Is it appropriate to include these in the 'grass' category? I have no expertise in this area, so my questions are based on logic rather than specific knowledge of this index. You should be certain that you have taken a valid approach, not just copied the approach of one other author who might have been wrong.

7) Goals and hypotheses
In the previous decision, I asked: How do your findings relate to diet changes in relation to the active growing season in other ungulates?
Your response: Although our study is not about diet change, we added some information about it in third paragraph.
The statement that your study is not about diet change does not match the organization of your manuscript. The first sentence of the Abstract and the first paragraph of the Introduction raise the issue of diet during seasons of food scarcity. This prominent position leads the reader to believe that it is the topic of your study. Your objectives specifically include this topic (L136,139,141). Therefore, it is important that you evaluate your findings in relation to seasons of higher availability, even if you do not have data on this season from your own study. Of course, this comparison must be done carefully because variables other than season might cause differences between your study and others.

More broadly, I have concerns about your hypotheses. You state that your hypothesis was that FHA would show equal preference for all species and categories (L136, 233-235). You do not provide any basis for this hypothesis. Indeed, this hypothesis cannot logically be correct. For example, if FHA prefer all species equally, then families with more species must make a larger contribution to the diet and therefore be unequal to families with fewer species. Similarly, for the other categories: if there are more dicot than moncot species and species are consumed equally, then there must be more fragments from dicots than from monocots. Furthermore, it is not clear that this hypothesis is compatible with your predictions about use of dicots and monocots (L136-141) in a season of lower availability. If equal use was truly a hypothesis that you wished to test before starting the study, you need to explain and justify it more fully. If this has the status of a 'null hypothesis' that you used to examine patterns in relative use, you should present it as a null hypothesis and be clear about what hypothesis, if any, you are testing wherever the topic arises (Introduction, Methods, Discussion).

8) Interpretation of results
I have some concerns about your interpretation of your findings:
i) It is important to remember that the absence of a significant difference between two groups does not allow you to conclude that they are the same. Lack of significance could result from small sample size or high variation. You have a logical error of this type in the Abstract (L36) and text (L244) and should check that you have not made similar statements elsewhere. You need to refer to the pattern as not significantly different from uniform, rather than stating that it is uniform.
ii) As I noted in my previous decision, a significant difference from a uniform distribution does not imply preference, because preference is a deviation from random choice (availability) which you did not measure. You made this error on L233. You refer to preference quite often in your literature review and should check that the authors really did assess preference and not simply a higher proportion of a certain type of food.

9) Use of English
Your revision has improved, but still contains errors. I did not carefully read the complete manuscript because it may change when you make corrections in response to my comments above. However, in examining the manuscript, I did note some examples where mistakes continued. This suggests that you still need to get a native English speaker with good writing skills to check the entire manuscript after you revise it. The lines where I noted problems (only a sample) are the following:
i) missing article (L59,68,101,240,253)
ii) missing hyphen (L101)
iii) missing word (L136,204)
iv) wrong spacing or use of hyphen (L138,139)
v) error in singular/plural (L248,254 note that dicotyledonous and monocotyledonous are adjectives and require a noun)
vi) word use (L313)

---

## Round 0.3 · Minor Revisions

I am returning your manuscript again because your resubmission was not adequate.

The rebuttal letter failed to mention points 3 and 4 of my decision.

Errors in English are still far too numerous. On pp. 4-7, I provided detailed comments indicating the numerous mistakes. On the other pages, I highlighted some of the problematic wording that I noticed on my first reading. This is not a complete list of the errors.

On L247, your p-value is probably not correct.

On L247-252, it seems likely that you indicated p < 0.05 when in fact the p-values are much lower (I suspect they are often < 0.001). You should give the lowest p-value that matches the chi square and df value. Stating p < 0.05 will imply to many readers that the p is between 0.05 and 0.01 rather than much lower.

Please note that because of the errors in your Version 2, I was unable to complete my review. It is possible that when I do so, other errors will be found. I strongly recommend that you undertake a careful review of the entire manuscript to reduce that possibility as much as possible.

Finally, I have a question about your Supplementary Table that might have implications for the Methods. I was curious about the frequent repetition of certain numbers (such as the 5 occurrences of 2.52) in Table 1. When I looked at the Supplementary Table to see how these came about, I was surprised to see that the totals for all 53 species and unidentified fragments were multiples of 5. This doesn’t make sense to me from my understanding of your methods. Could you please explain how this pattern could occur and whether any clarification of the methods is required?

---

## Round 0.4 · accepted · Accept

The manuscript is now ready for publication. I have provided an annotated pdf with some minor corrections that we agreed to by separate email correspondence. These can be incorporated while in Production

#